# Months-long seismicity transients preceding the 2023 $M_W$ 7.8 Kahramanmaraş earthquake, Türkiye

G. Kwiatek [1], P. Martínez-Garzón [1] ✉, D. Becker [1], G. Dresen[1,2], F. Cotton [1,2], G. C. Beroza [3], D. Acarel [4], S. Ergintav [5] & M. Bohnhoff [1,6]

Short term prediction of earthquake magnitude, time, and location is currently not possible. In some cases, however, documented observations have been retrospectively considered as precursory. Here we present seismicity transients starting approx. 8 months before the 2023 $M_W$ 7.8 Kahramanmaraş earthquake on the East Anatolian Fault Zone. Seismicity is composed of isolated spatio-temporal clusters within 65 km of future epicentre, displaying non-Poissonian inter-event time statistics, magnitude correlations and low Gutenberg-Richter $b$-values. Local comparable seismic transients have not been observed, at least since 2014. Close to epicentre and during the weeks prior to its rupture, only scarce seismic activity was observed. The trends of seismic preparatory attributes for this earthquake follow those previously documented in both laboratory stick-slip tests and numerical models of heterogeneous earthquake rupture affecting multiple fault segments. More comprehensive earthquake monitoring together with long-term seismic records may facilitate recognizing earthquake preparation processes from other regional deformation transients.

Individual earthquakes cannot be predicted in a deterministic way despite the urgent social-economic need to warn people and protect critical infrastructure. In some cases, however, the processes leading to the nucleation of an earthquake may have an extended duration, i.e. months-to-years, that can be monitored, and potentially recognised.

Earthquake foreshocks are the most relevant detectable earthquake precursors, but their identification as such is typically only possible once the mainshock has occurred. Their predictive power may be tied to whether the foreshock collective behaviour can be uniquely attributed to loading by aseismic slip and/or fluid flow, meaning that the foreshocks act as passive tracers of the preparatory process[1]. Some interplate earthquakes have been observed to be preceded by increased seismic and aseismic slip in the region leading up to the mainshock[2]. This has been particularly well documented in several subduction zone megathrust earthquakes[3–5]. An alternative view is that foreshocks occur as part of a cascade of failures, such that the large earthquake that occurs after them is governed by the proximity of a fault to failure in a large event[6].

On the decadal time scale, during the inter-seismic period, increasing levels of background seismic activity may represent enhanced damage generation over a broader area hosting a notable future earthquake[7]. In some large earthquakes on plate-bounding transform faults, such as the 1992 $M_W$ 7.3 Landers, the 1999 $M_W$ 7.1 Hector Mine and the 2019 $M_W$ 7.1 Ridgecrest events, spatio-temporal localisation of the seismicity towards the main fault traces (the eventual rupture plane) has been observed over time scales of 2-3 years before the mainshock[8], promoting the interaction

[1]Helmholtz Centre Potsdam GFZ German Research Centre for Geosciences, Potsdam, Germany. [2]Institute of Geosciences, University of Potsdam, Potsdam, Germany. [3]Department of Geophysics, Stanford University, Stanford, CA, USA. [4]Institute of Earth and Marine Sciences, Gebze Technical University, Gebze-Kocaeli, Turkey. [5]Department of Geodesy, Kandilli Observatory and Earthquake Research Institute, Boğaziçi University, Çengelköy-Istanbul, Turkey. [6]Free University Berlin, Institute of Geological Sciences, Berlin, Germany. ✉e-mail: patricia@gfz-potsdam.de

between seismicity and the coalescence of fault branches and fractures framing the principal slip zone activated in future large earthquakes. The ultimate phase just before rupture may include the occurrence of slow-slip transients and/or foreshocks at multiple spatial and temporal scales[9]. Tracking these processes and identifying slow-slip events or small earthquakes as indicators of an upcoming large earthquake, however, remains a challenge. This is the case as the long recurrence times of large earthquakes of typically a century or more only provide a limited number of examples since the onset of instrumental seismology. An additional challenge is that analysing long-term deformation records, seismic and/or aseismic deformation transients can also occur without resulting in a major earthquake[10].

We here document seismic signatures on the Pazarcık segment of the East Anatolian Fault Zone (EAFZ) that preceded the rupture of a major earthquake over months before the mainshock. A densification of seismic monitoring in the epicentral area in 2014[11] substantially improved the earthquake detection threshold, which enabled the search for seismicity transients in greater detail. We additionally access continuous recordings from more than 40 stations in the area operated by AFAD (Disaster and Emergency Management Authority, Ankara) and KOERI (Kandilli Observatory and Earthquake Research Institute, Boğaziçi University, Istanbul) agencies, to recover a high-resolution view of the seismicity during the preceding weeks. The observed seismicity transients suggest an extended earthquake preparation process starting in June 2022 and lasting for approximately 8 months prior to the occurrence of the $M_W$ 7.8 Kahramanmaraş earthquake, including accelerating seismic activity, non-Poissonian inter-event time statistics and magnitude distribution in time, as well as low Gutenberg-Richter $b$-values. Close to the mainshock epicentre and during the weeks prior to the mainshock, only scarce seismic activity is observed. These observations suggest a different initiation mechanism compared to the cascade of close (<200 m) foreshocks observed before other large strike-slip earthquakes including the $M_W$ 7.6 1999 Izmit event.

## Results

### The February 6th, 2023 Kahramanmaraş earthquake

On February 6th, 2023 at 01:17 UTC, a devastating $M_W$ 7.8 earthquake struck the Pazarcık segment of the EAFZ bounding the Arabian and Anatolian tectonic plates[12] (Fig. 1). The epicentral area was located in southeastern Türkiye, in a broad region including the municipalities of Gaziantep and Hatay, home to more than 2 million inhabitants. Immediate aftershocks included a $M_W$ 6.7 only 10 min after the mainshock as well as several tens of $M_W > 4$ aftershocks. Nine hours later, a $M_W$ 7.5 earthquake occurred about 90 km to the NNW of the initial mainshock. This event was likely triggered by stress redistribution induced by the $M_W$ 7.8 earthquake[13]. The mainshock ruptured the entire seismogenic crust reaching to the surface and covered a length of about 500 km, activating several fault branches and propagating across several step-overs. The elevated ground shaking together with the abundant aftershock seismicity left a toll of at least 50,399 and 8476 casualties in south-eastern Türkiye and northern Syria, respectively (USGS 2023, https://earthquake.usgs.gov/earthquakes/eventpage/us6000jllz/impact, last accessed 05/09/2023). Although the earthquake ruptured both the Pazarcık and Amanos segments of the EAFZ, the nucleation point was located off the main fault branch on a splay fault to the east. The rupture then propagated dynamically onto the main fault trace (Fig. 1). This behaviour is similar to the rupture of some other major earthquakes on transform faults such as the $M_W$ 7.9 Denali/Alaska[14] and the $M_W$ 7.8 Kaikoura/New Zealand[15] earthquakes.

The EAFZ is a left-lateral strike-slip fault that forms the boundary between the Anatolian and Arabian tectonic plates[16]. It has a length of ~750 km along strike between the triple junction connecting the EAFZ, the Dead Sea Transform and the Cyprus Arc in the SW and the Karlıova junction connecting the EAFZ with the North Anatolian Fault Zone (NAFZ) in the NE. Representing the second largest fault zone in Türkiye after the NAFZ, the seismic activity on the EAFZ has been monitored by the national agencies AFAD and KOERI with increasing efforts to assess the seismic hazard and risk of the region. Geodetic and geological studies indicate that the slip rates along the main fault

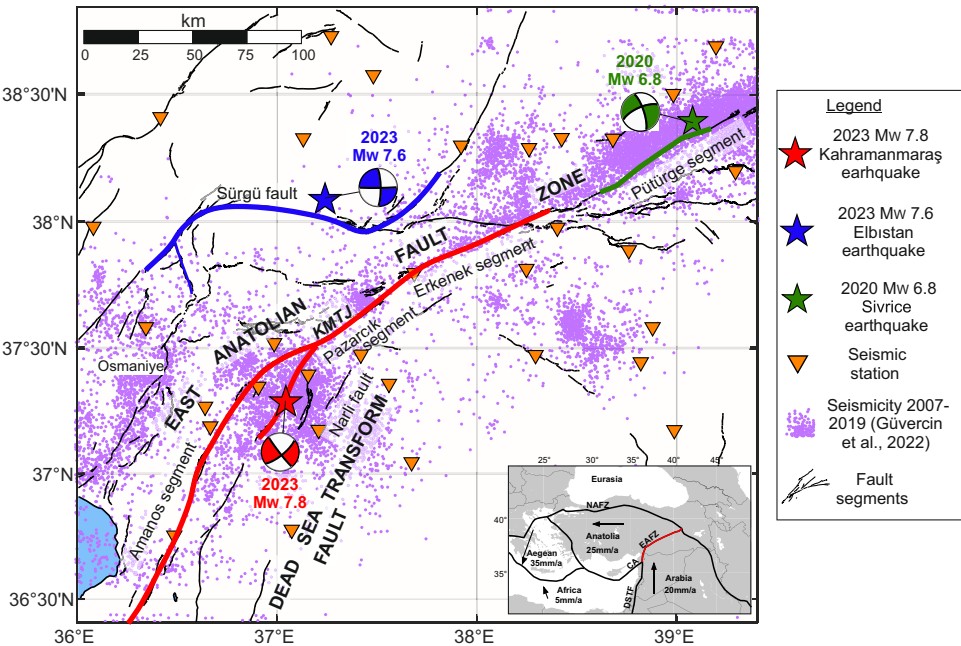

**Fig. 1 | Regional seismicity before the Kahramanmaraş earthquake.** Fault traces and focal mechanisms associated with the 2023 $M_W$ 7.8 Kahramanmaraş and 2023 $M_W$ 7.5 Elbistan earthquakes are shown in red and blue, respectively. Fault trace, epicentral location and rupture mechanism of the 2020 $M_W$ 6.8 Elazig/Sivrice earthquake is also shown in green for comparison. Surface ruptures as reported by USGS map and ref. 34. The focal mechanisms display the estimated moment tensor for these earthquakes from GEOFON (lower hemisphere projection). Inset shows the major tectonic plates and politic boundaries in this region. Pink dots represent the seismicity during 2007–2019[21]. Fault surface traces are from[62].

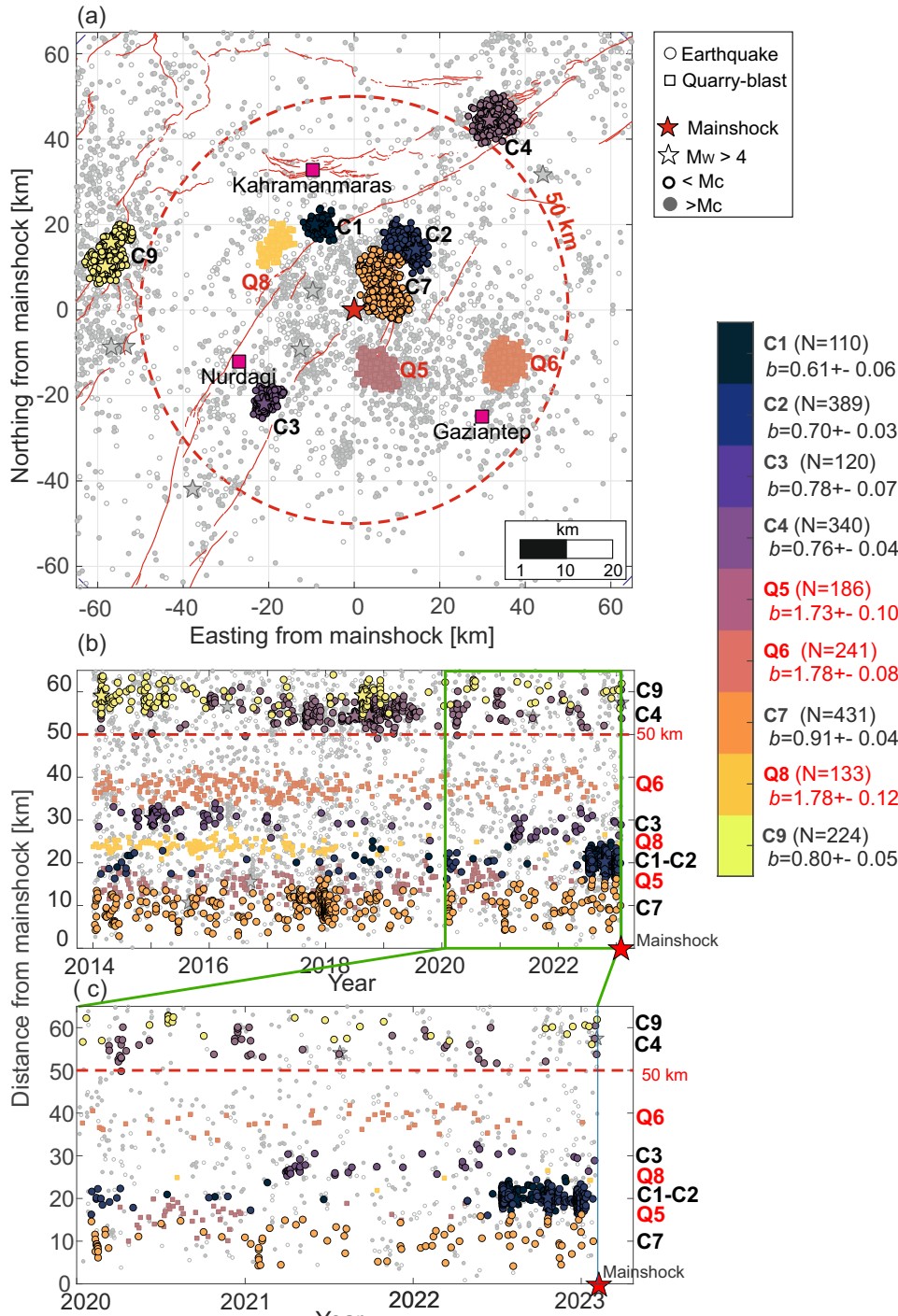

**Fig. 2 | Spatio-temporal evolution of seismicity before the $M_W$ 7.8 mainshock. a** Clustered seismic activity (C1-C3, C4, C7, C9) and quarry blasts (Q5, Q6, Q8, labelled in red) are visible within a 65 km radius surrounding the future $M_W$ 7.8 epicentre. Seismicity clusters display remarkably lower *b*-values compared to the quarry blasts and non-Poissonian inter-event time statistics (Supplementary Fig. 6). Clusters C1 and C2 hosting the largest 4 foreshocks and emerging within 8 months prior to the mainshock display the lowest *b*-values, non-Poissonian inter-event time statistics and (in case of C1) magnitude correlations (cf. Supplementary Figs. 5 and 6). **b** Temporal evolution of seismicity and quarry blast clusters since 2014 with respect to epicentral distance from the mainshock nucleation point. **c** Same as (**b**) but zooming-in the time period starting 2020.

trace of the EAFZ and its secondary branches decrease from ~10 mm/yr near the Karlıova triple junction in the NE down to ~4 mm/yr near Kahramanmaraş junction[17]. The centuries-long historical earthquake records of Anatolia indicate that large earthquakes have previously occurred on the EAFZ. On the Pazarcık segment where the 2023 $M_W$ 7.8 Kahramanmaraş earthquake occurred, historical reports document an earthquake of similar magnitude in the year 1114[18,19]. The last

large (*M* 7.0) earthquake occurred in 1795 on this segment[20]. Toward the NE, the last large earthquake on the EAFZ was the 2020 $M_W$ 6.8 Sivrice earthquake, which ruptured about 45 km of the Pütürge segment[21] (Fig. 1). The EAFZ is a strongly segmented fault zone characterised by dominantly NE-SW and E-W striking fault traces[22] displaying an evolving network with varying width and on- and off-fault deformation[23–25].

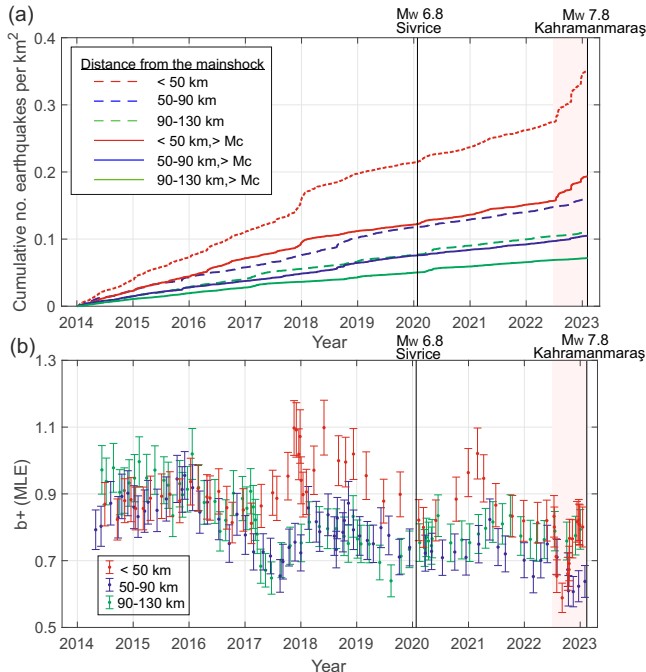

**Fig. 3 | Temporal evolution of cumulative seismic activity and b-values at different distances from $M_W$ 7.8 epicentre. a** Cumulative seismic activity per km² was calculated for different epicentral distances from the $M_W$ 7.8 mainshock (excluding quarry blast clusters Q5, Q6 and Q8). Dashed and solid lines represent the entire and complete catalogues ($M_C > 1.5$) in each zone, respectively. The seismicity rates accelerated within the last 8 months before the mainshock within 50 km radius mostly due to clusters C1 and C2 (Supplementary Fig. 1). **b** The b-value (with error bars showing 95% c.i.) displays a transient significant decrease within a 50 km radius around the epicentre and a 8-month period before the mainshock that is primarily attributed to C1 and C2, and a subsequent recovery is visible. This transient decrease is overlain with a long-term decrease in the b-values in the broader region $50 < R < 90$ km. Transient drops in b-value observed in 2017 and 2019 for $50 < R < 130$ km are attributed to scattered seismic activity in the Osmaniye region (see Fig. 1 for location) that do not display comparable clustering.

## Months-long activation surrounding the nucleation region

To understand how the seismicity around the $M_W$ 7.8 Kahramanmaraş earthquake rupture evolved over the months and years prior to the rupture, we analysed the spatio-temporal evolution and statistical properties of earthquake activity using the AFAD seismicity catalogue for a broad region with coordinates Latitude [36°, 39°] and Longitude [35°, 39°]. We started our analysis in January 2014 coinciding with the effective incorporation of seismic stations to the generation of the local seismicity catalogue by AFAD that visibly improves the consistency of magnitude estimates since the selected time. The input catalogue initially contained $N = 9483$ earthquakes that occurred within a radius of $R < 130$ km from the future mainshock epicentre (following AFAD epicentre location assessment) and $N = 5041$ events above the local completeness magnitude $M_C = 1.5$, estimated using the goodness-of-fit method[26].

The Pazarcık segment and secondary splay faults displayed abundant seismicity. This includes nine automatically selected earthquake clusters located on and off the main fault trace (see 'Methods' section). These clusters were identified from seismicity located at epicentral distances $R$ not exceeding 65 km from the epicentre of the $M_W$ 7.8 earthquake (Fig. 2). Detailed analysis of the initial catalogue revealed that clusters Q5, Q6 and Q8 corresponded to quarry-blast activity. These findings were based on the following: (I) quarry blast clusters showed a narrow-magnitude band (Supplementary Fig. 1a), (II) quarry blast clusters displayed non-uniform hourly distributions of events (Supplementary Fig. 1b), (III) event locations coincided with

quarry blasts reported in the KOERI catalogue (50-95% events attributed to blasts) and with (IV) quarry sites located nearby, as observed in the digital terrain maps (OpenStreetMap service). These clusters were not interpreted further. Following criteria I–IV, the remaining clusters C1-C4, C7 and C9 were considered of tectonic origin with a residual no. of blasts (<3%) reported for C1-C4, C9, and ~10% of blasts for C7. The higher percentage of blasts reported for C7 was due to the proximity of several quarries. However, as the remaining features (I, II, III) were not observed (cf. Supplementary Fig. 1c, d), C7 was considered dominantly of tectonic origin.

The nucleation region ($R < 10$ km) hosted spatially dispersed seismicity ($N = 369$, $M_{C,R<10} = 1.25$, $n(M > M_{C,R<10}) = 261$) with $b = 0.9 \pm 0.1$ and stationary seismicity rates in the 9 years preceding the mainshock. Quarry blast clusters Q5, Q6 and Q8 located within the region ($R < 50$ km) also displayed quasi-stationary rates throughout the analysed time period (Supplementary Figs. 2 and 3). Clusters C3 and C4 located closer to the main EAFZ branch displayed prominent seismicity occurring in mid-2021 and through 2018–2019, respectively. Abundant activity occurred in C7 between Jul-2017 and Mar-2018. This was composed of comparably small events (Supplementary Figs. 3, 1a) located closest to the future $M_W$ 7.8 earthquake epicentre with $b = 0.9 \pm 0.1$, that is similar to that observed in the region ($R < 90$ km, $b = 0.9 \pm 0.1$). Of particular importance are the on-fault seismicity clusters C1 and C2 that started nearly simultaneously in July 2022 on the main EAFZ and the splay fault hosting the future $M_W$ 7.8 epicentre, respectively. They contained distinct event sequences with pronounced temporal clustering and two $M_W > 4$ earthquakes each (stars in Fig. 2a, b). These $M_W > 4$ earthquakes resulted in prominent aftershock activity and subsequent sequences which lasted until the occurrence of the $M_W$ 7.8 mainshock (Supplementary Fig. 2). Until July 2022, regions covered by future clusters C1 and C2 exhibited minor seismic activity. The remaining earthquake clusters displayed stationary seismicity rates in the 8 months preceding the mainshock, meaning that C1 and C2 solely contributed to the sudden acceleration in seismicity rates (red line in Fig. 3a) and moment rates (Supplementary Fig. 4) in this time period. This acceleration is not observed at distances $R > 50$ km from the $M_W$ 7.8 epicentre (at least up to $R < 130$ km from the future earthquake epicentre, see blue and green lines in Fig. 3a), nor is it observed in the future nucleation area ($R < 10$ km). In the last months preceding the mainshock, the region surrounding the future earthquake up to $R < 50$ km was characterised by a transient b-value decrease related to the activation of C1 and C2 clusters, overprinting the b-value decrease observed in the broader region ($R < 90$ km) since mid-2021 (Fig. 3b). Transient b-value drops were also observed in 2017 and 2019 for the region within $50 < R < 90$ km from the epicentre. However, these transient drops were attributed to spatially scattered seismic activity in the Osmaniye region (cf. Fig. 1). Overall, no similar and concurrent increase in the seismicity rates and decrease in the b-value attributed to distinct spatio-temporal clusters can be observed in the region during the preceding nine years. However, we note that at the beginning of 2018 cluster C7 displayed an increase in seismicity rate that did not result in a large earthquake rupture. This rate increase is comparable to that observed before the $M_W$ 7.8 earthquake, but shows no b-value decrease. Interestingly, after the C7 activation, the seismicity rates decreased in the region with $R < 50$ km from the future $M_W$ 7.8 epicentre (Figs. 2b and 3a). At the same time, the seismic rates in a broader region ($50 < R < 130$ km) remained stationary throughout the whole analysed time period. Finally, in January 2020, the $M_W$ 6.8 Sivrice/Elazig earthquake occurred on the NE portion of the EAFZ. No permanent changes in the seismicity rates from the study region are observed after this earthquake within the broader region $R < 130$ km (Figs. 2c and 3a).

Quarry blast clusters Q5, Q6 and Q8 clearly exhibited the highest b-values ($b = 1.73 \pm 0.10$, $b = 1.78 \pm 0.08$, $b = 1.78 \pm 0.13$ for Q5, Q6 and Q8, respectively, Fig. 2a). Clusters C3, C4 and C7 present b-values relatively close to that reported for the Pazarcık and Amanos fault segments

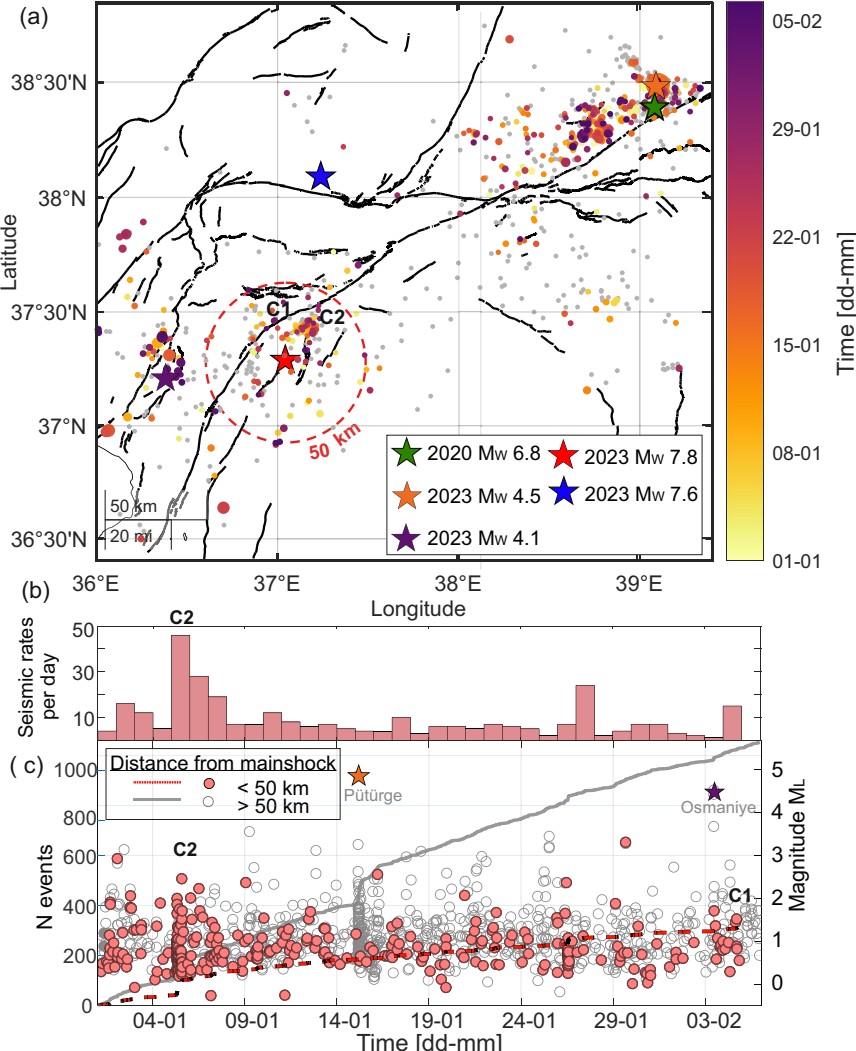

**Fig. 4 | Spatial and temporal evolution of seismicity from January 1st, 2023 up to the $M_W$ 7.8 mainshock, from high-resolution seismicity catalogue.**
**a** Coloured dots indicate single event NLLoc locations with major error ellipsoid half-axis <10 km and grey dots indicate all other events. The locations were derived from the high-resolution catalogue spanning from Jan 1st, 2023 to the mainshock time. $M_W$ 7.8 Kahramanmaraş and $M_W$ 7.6 Elbistan earthquakes are shown with red and blue stars, respectively, whereas the green star marks the epicentral location of the 2020 Sivrice/Elazig earthquake. The orange and purple stars highlight the $M_W$

4.5 Pütürge and $M_W$ 4.1 Osmaniye events discussed in the text. **b** Daily seismicity rates in a circular region within a 50 km radius from the mainshock epicentre (dashed red circle in (**a**)). **c** Cumulative number of events for the entire area shown in (**a**) and within $R < 50$ km around the mainshock (solid grey and dashed red lines, respectively). Circle symbols show the magnitude of events over time covering the entire area shown in (**a**) and for the region within $R = 50$ km radius around the mainshock (grey bordered and red filled, respectively). Surface fault traces are from[62].

during the interseismic period ($b = 0.9$, see[25]). However, the features of these clusters are significantly different from that observed for clusters C1 and C2 that directly precede the mainshock. The latter two clusters display very low $b$-values down to $b = 0.61 \pm 0.06$ and $b = 0.7 \pm 0.03$, respectively, indicating the presence of a highly stressed region[27] or damage concentration[28] to the north of the future nucleation zone in the last months preceding the earthquake. In addition, cluster C1 displays a statistically significant deviation from a random (Poissonian) magnitude distribution in time (i.e. magnitude correlations) indicating some level of interaction between subsequent events (Supplementary Fig. 5a). Finally, all tectonic clusters display presence of aftershock sequences leading to statistically significant temporal clustering of the seismicity (Supplementary Fig. 6).

**The lead up to the mainshock**
To identify potential immediate foreshocks around the $M_W$ 7.8 Kahramanmaraş earthquake, we generated an enhanced seismicity catalogue lowering the magnitude detection threshold around the $M_W$ 7.8

Kahramanmaraş and $M_W$ 7.5 Elbistan earthquakes for a time period starting from January 1st up to the mainshock on February 6th 2023. We used an Artificial Intelligence (AI)-based workflow[29–31] to generate a catalogue of earthquake detections and locations. The initial catalogue locations were refined using the absolute (NLLoc[32]) and, for selected regions, relative (HypoDD[33]) location techniques (see Methods section). This led to a single event NLLoc catalogue of 1055 located events that occurred before the mainshock in the study area. Excluding the events located within the Q5, Q6 and Q8 areas previously identified as quarry blasts left 1029 events representing earthquakes. The catalogue has a local magnitude of completeness of $M_C = 0.48$ in the 50 km radius around the mainshock and records a total of 252 events above $M_C$ since January 1st 2023. For comparison, the AFAD catalogue for the same region displays a $M_C = 1.45$, and a total of 22 events above $M_C$ within 50 km radius from the mainshock during the same time period.

Focusing on the region within $R < 50$ km of the epicentre of the $M_W$ 7.8 earthquake, the C2 area activating the northern end of the secondary segment hosted 151 events since Jan 1st, becoming most active

on Jan 5th and 26th (Fig. 4 and Supplementary Fig. 7). Activity within the first half of January is clearly aligned along the strike of the future initial rupture plane, showing a migration towards the NNE as well as a slight deepening (Supplementary Fig. 8). This activity continued until only six days before the mainshock. The C1 region, directly on the main fault trace of the Pazarcık segment of the EAFZ, hosted 7 microseismic events within this time period, the last one being a $M_L$ 1 event 43 h before the mainshock. The junction between the Pazarcık and Erkenek segments hosted very few seismic events during the five weeks before the mainshock rupture, in agreement with the low seismicity rates observed during the interseismic period (Fig. 4, see also[25]). Interestingly, co-seismic slip during the mainshock was the largest around this segment[34], signifying a large stress accumulation on this segment during the interseismic period. In contrast, the Pütürge segment of the EAFZ displayed abundant seismicity during this time period, including a $M_W$ 4.5 earthquake and its corresponding aftershock sequence from January 15th (orange star in Fig. 4). This seismicity could have contributed to the gradual unlocking of this fault segment. It is notable that the $M_W$ 7.8 rupture approximately stopped at the edges of this area. That segment of the fault has continuous seismic activity and coupling is thought to be either weak or heterogeneous[21].

The week before the mainshock was characterised by relatively low seismicity rates around the nucleation area of the $M_W$ 7.8 event (Fig. 4 and Supplementary Fig. 7). On February 3rd a $M_W$ 4.1 event ruptured on a sub-parallel branch near Osmaniye, about 60 km away from the future mainshock. The seismicity following this event was located both in the vicinity of the $M_W$ 4.1 event, and away from it, including the main branch of the EAFZ and the western secondary fault where the mainshock nucleated.

## Discussion

Conceptual models such as the cascade, preslip and progressive localisation have been developed describing preparation and nucleation processes of earthquakes[6,7]. The models reflect the richness of the observed deformation behaviour during run-up to failure that is caused by varying degrees of structural complexity, segmentation of faults and heterogeneous stresses, as also demonstrated in a plethora of laboratory and numerical modelling studies[35–37]. The EAFZ represents a strongly segmented fault zone typically hosting an interplay of main strand left-lateral strike-slip mechanisms and subordinate normal- and thrust faulting mostly located off-fault[38]. Episodic interaction between main and secondary fault branches was previously documented from space-time evolution of seismic clusters[25,38].

Our observations of the seismicity evolution prior to the mainshock show similarities with long-lasting preparation processes of large events on other large and complex faults such as the $M > 7$ Landers 1992 and Ridgecrest 2019 earthquakes[9,39]. For these continental strike-slip earthquakes, foreshocks from multiple event clusters on different nearby faults are part of a regional shear localisation process that eventually leads to the generation of a large earthquake. In the Pazarcık segment of the EAFZ, the seismicity indicates a progressive evolution that is not limited to a certain pre-existing fault, but is instead distributed in the form of clusters in a rather large volume surrounding the future epicentral zone.

In the sequence studied here the seismicity did not show a progressive localisation towards the future hypocentre, and none of the identified seismicity clusters corresponded to its nucleation zone ($R < 10$ km). Moreover, with the enhanced seismicity catalogue, we did not find evidence of a cascade process due to immediate foreshocks (with $M_W > 0.48$) located at distances less than 500 m from the epicentre as observed before the 1999 $M_W$ 7.4 Izmit earthquake[40] or the 1999 $M_W$ 7.1 Hector Mine[41]. This difference in behaviour supports the diversity of the initiation process of large earthquakes. The 1999 Izmit earthquake occurred on a structurally mature fault with a cumulative offset of up to 90 km, likely resulting in progressive smoothing of the fault plane. The south-western portion of the EAFZ where the 2023 earthquake initiated is a comparatively less mature tectonic structure, accounting for only up to 27–30 km cumulative offset and numerous secondary fault systems[38,42,43].

We observe a substantial increase in the seismic event rates surrounding the $M_W$ 7.8 epicentre by a factor >3 after July 2022 (Fig. 3a). A close correlation of aseismic slip and enhanced seismic activity has been observed prior to some large earthquakes on transform faults and subduction zones[44]. Although aseismic slip may have contributed to the enhanced deformation rates in the surrounding area, as of now there was no clear GNSS signature or other indications from seismicity.

The presence of localised seismic activity clusters bears similarities to observations of seismicity ahead of large stick-slip events on rough laboratory faults that emphasise the importance of fault heterogeneity/complexity in the preparation process[36,45]. Acoustic emission (AE) activity ($M_W$ −10 to −7) rates prior to system-wide stick-slip failure often show an exponential increase, especially for smooth faults[36]. However, rough laboratory faults that likely are more representative of complex faults in nature[36,46] display persistent and/or transient local seismicity clusters (reflecting asperities evolution) at high levels of stress. As in our field observations, laboratory AE clusters often result in complex accelerating and decelerating seismic activity (not necessarily leading to a major event), varying $b$-values reflecting transient and local stress changes and damage localisation[35,36,47]. In these experiments the evolution of clusters progresses, collectively preparing a wider fault area for large failure events[47,48]. This suggests, the nucleation of a large earthquake somewhere along rough heterogeneous faults is a statistical event, conditioned by an intrinsically complex failure process of individual asperities of various sizes and strengths[48].

Our observations illustrate the challenge of detecting the preparation and initiation phase of large earthquakes. The seismic rupture occurred on a fault and in a region identified to have a very high seismic hazard potential. Consequently, a M 7–8-type earthquake (in terms of rupture size and location) was present in the probabilistic seismic hazard models published before the earthquake[49]. The initiation phase, on the other hand, takes place on a secondary fault that has received little attention. Our study shows that a full understanding of the initiation phenomenon, necessary for any future development of warnings, would require a greater densification of local and regional seismological and geodetic networks to reduce or even close the observational gap between the laboratory (where such processes can be observed) and the field. This seismic sequence, in addition to the Kaikoura, Denali and Ridgecrest sequences, shows that this densification should not be done only in the immediate vicinity of major faults, but on a regional scale that includes secondary faults. Several dense, regional and multi-parametric networks have been developed in the last few decades (e.g., KiK-net, Hi-net, USArray, ChinArray and near-fault observatories[50–52]). These efforts must be intensified and extended to more major fault systems[49] to improve the chances of understanding the predictability of large earthquakes.

Our results add to the accumulating evidence that at least some large earthquakes display a monitorable preparation phase that bears some similarity to laboratory and theoretical models of the failure process. However, the variability of apparent earthquake nucleation processes observed for different events, the difficulty of distinguishing preparatory processes from other deformation transients that do not lead to major earthquakes, the participation of secondary faults, and an unknown false alarm rate, all suggest that with our current state of knowledge, intermediate-term earthquake warning—if possible—still lies in seismology's future.

## Methods

### Long-term catalogue processing and cluster selection
In the analysis we used a seismic catalogue provided by AFAD Disaster and Emergency Management Authority in Turkey[53] between January

1st, 2014 and the occurrence of the $M_W$ 7.8 mainshock. The selection of the start time for the catalogue was chosen because earlier times were affected by strong changes in the regional network[11] and hence, by different earthquake detection thresholds. The catalogue was constrained spatially to seismic events located within 130 km epicentral distance from the $M_W$ 7.8 mainshock epicentre provided in the same catalogue (Origin time 2023-02-06T01:17:32 UCS, Latitude 37.288°, Longitude 37.043°, coordinate (0 km, 0 km) in Fig. 2). The final, spatially and temporally constrained catalogue contained $N = 9483$ events.

To calculate cumulative seismic moment release (Supplementary Fig. 4) we used local-to-moment magnitude conversion formula:[54]

$$M_W = 1.053 M_L - 0.105, \quad (1)$$

and converted the resulting moment magnitude to the seismic moment using standard relation[55]:

$$M_0 = 10^{1.5 M_W + 9.1} [\text{Nm}]. \quad (2)$$

Nine spatio-temporal clusters of seismicity were selected in the area surrounding the mainshock ($R < 65$ km) using DBSCAN algorithm[56] assuming key parameter epsilon=2.5 km and minimum number of events in the cluster $N = 40$. We then investigate differences in seismicity rates, seismic energy release, $b$-value, magnitude correlations and interevent time statistics for each cluster.

The analysis of the seismicity in each cluster was performed assuming (when necessary) the conservative magnitude of completeness ($M_C = 1.5$, as estimated for the broad region $R < 130$ km surrounding the future $M_W$ 7.8 epicentre). The $b$-value calculations for each selected cluster of seismicity followed the $b$-positive method[57] via the maximum likelihood estimator. The $b$-positive methodology suppresses the potential bias from short-term catalogue incompleteness. The uncertainties (95% c.i.) in $b$-value were estimated using bootstrap resampling[57]. Visual inspection of probability density functions (PDFs) of the magnitude and origin times (Supplementary Fig. 1), proximity to the quarry sites identified using OpenStreetMaps service, as well as the proximity to blasts identified in the smaller KOERI catalogue revealed that clusters Q5, Q6 and Q8 contained a significant portion of events related to quarry blasts. Therefore, these clusters were further not interpreted and the associated events were removed from analysis.

## Magnitude of completeness and $b$-value

For the selected portion of the seismicity catalogue ($R < 130$ km, AFAD catalogue since 2014), magnitude of completeness $M_C$ was calculated using the goodness-of-fit method[26]. The $M_C$ was selected in such a way that 95% of earthquakes forming the selected catalogue follow the Gutenberg–Richter power law. The $b$-value itself was calculated using the $b$-positive method[57]. For the initially selected catalogue of $N = 9483$ earthquakes, we obtained $b = 0.9 \pm 0.1$ with $N = 5041$ events above $M_C = 1.5$.

To investigate the temporal evolution of the $b$-value presented in Fig. 3b in different zones surrounding the earthquake epicentre ($R < 50$ km, $50 < R < 90$ km, and $90 < R < 130$ km), we used a moving event window of 120 events with a step of 15–20 events and attributed each calculated $b$-value to the origin time of the last event in each window. Estimation of $b$-value in each window followed as well the $b$-positive method[57]. Before calculation, we removed seismicity associated with clusters Q5, Q6 and Q8 containing quarry blasts. Two more clusters detected in a broader region (up to $R < 130$ km) related to quarry blasts, as evidenced from KOERI catalogue, were removed as well.

## Magnitude correlations

For each cluster C1–C9 and Q5, Q6 and Q8, we extracted events above $M_C$ and tested for magnitude correlations between consecutive events

forming the cluster. Magnitude correlations between events would indicate that the events are not randomly drawn from the G-R distribution as expected from a Poissonian process. The input to the statistics is the time-ordered sequence of magnitudes $[M_i]$ above $M_C$:

$$\mathbf{\Delta M} = [\Delta M_i] = M_{i+1} - M_i. \quad (3)$$

Following ref. 58, the PDF of empirical magnitude difference data, $p(\mathbf{\Delta M})$ is correlated, if it significantly deviates from the distribution of magnitude differences containing uncorrelated magnitudes, $p(\mathbf{\Delta M})$. Such a distribution can be constructed multiple times by considering $\Delta M_i^\star = M_{i^*} - M_i$, where $M_{i^*}$ is a magnitude randomly drawn from the original catalogue. The difference between the cumulative distribution function (CDF) calculated from empirical data and multiple realisations of the CDFs calculated from perturbed vectors of magnitudes is calculated as:

$$\delta p(\mathbf{\Delta M}) = p(\mathbf{\Delta M} < \Delta m) - p(\mathbf{\Delta M} < \Delta m), \quad (4)$$

(Supplementary Fig. 4). When magnitudes in the original catalogue are correlated, $\delta p(\mathbf{\Delta M})$ should significantly deviate from 0 for all considered $\Delta m$. In this case, the catalogue magnitudes are not behaving as randomly drawn from a Gutenberg–Richter relation. Deviation from random distribution of magnitudes in time may indicate the existence of local-in-time accelerated seismic release (e.g. in earthquake preparatory phase) or deceleration (e.g. as in aftershock sequences). When these processes occur in spatially clustered sequences, they may be indicative of earthquake interactions and stress transfer.

## Interevent time ratio

To calculate interevent time ratio, we followed ref. 59 where the temporally ordered seismicity catalogue $\mathbf{T} = [T_i] = T_{i+1} - T_i$ is used to calculate the following statistics:

$$\mathbf{R} = [R_i] = (T_{i+1} - T_i)/(T_{i+1} - T_{i-1}). \quad (5)$$

For a stationary or weakly time-varying Poisson process, the PDF of interevent time ratios, $p(R)$ is expected to display a uniform distribution in interval (0,1). Clustering and anti-clustering are expressed by significant peaks of the $p(\mathbf{R})$ close by $R = 0$ and $R = 1$, respectively. The statistical significance of this statistic can be measured by comparing observed $p(\mathbf{R})$ to that built upon the data sample randomly distributed over time that follows the Poisson process[58,59]. Similarly to the magnitude correlations, transient temporal clustering deviating from that expected for a quasi-stationary Poissonian process indicates significant acceleration or deceleration of seismicity. If such a process occurs in spatially clustered seismicity, it may be indicative of earthquake interactions and stress transfer.

## High-resolution seismicity catalogue

We processed continuous waveform recordings from 40 regional seismic stations (33 belonging to the Turkish National Seismic Network[53] and 7 to the Kandilli Observatory Network[60], see Fig. 1 and Supplementary Fig. 9). We covered the time period from January 1st 2023 up to the mainshock on February 6th, 2023 at 01:17 h UTC time.

We detected P- and S-wave onset times embedded in the continuous recordings applying the supervised AI method PhaseNet[29] trained on the seismicity database from northern California. This method has been shown to improve the detection process especially for small earthquakes[29]. The P- and S-picks were associated with seismic events using the unsupervised technique GaMMA[30]. To classify an event as an earthquake, we require a minimum of 6 picks (either P and/ or S). The picks were spatially and temporally clustered using DBSCAN[56]. This results in a catalogue of 1097 possible seismic events

with a total of 5868 P- and 4960 S-picks. We calculated new event locations by employing the probabilistic location software NLLoc[32,61] while using the regional, 1-D velocity model from [25]. The search area encompassed a 340 km × 400 km × 82 km volume with its lower left corner positioned at 36.0 N and 35.4 E. Based on the event origin times obtained from NLLoc we visually investigated all events with an origin time difference of less than 5 s for possible duplicate events. Such duplicates can be created by GaMMA by splitting P- and S-picks belonging to the same event into separate events. All identified duplicates were merged into a single event and relocated. This resulted in a catalogue of 1055 events with absolute locations and median errors in the x-, y- and z-directions of 4.8 km, 5.5 km and 6.5 km, respectively (Fig. 4 and Supplementary Fig. 7). The dataset is available as a part of Supplementary Information.

## Relative event relocation

We performed a relative relocation of the events occurring in the vicinity of cluster C2 during the beginning of 2023 using cross-correlation and catalogue differential times at all available common stations using the software hypoDD[33]. For a cross-correlation time (either P or S) to be used in the relocation procedure, we required a normalised cross-correlation coefficient of 0.6 or larger. To obtain cross-correlation differential times, we filtered waveforms between 2 and 10 Hz, and used 1 s and 2 s long-time windows centred at the P- and S-onset times, respectively. We combined these data with the differential travel times obtained from the automatic picks and ran hypoDD in singular value decomposition (SVD) mode to obtain the relocated event hypocentres. To link events for the relative relocation, we require at least 8 catalogue and 8 cross-correlation differential times for an event combination together with a maximum inter-event separation of 20 km from the single event localisations and a maximum station distance to the cluster centroid of 100 km. From a total of 89 initially selected events, 86 remain after relative relocation (Supplementary Fig. 8). The formal median errors of these events in x-, y- and z-directions are 18 m, 35 m and 57 m, respectively.

## Data availability

Waveform data used to create the enhanced seismicity catalogue was acquired from the Turkish National Seismic Network[53] and Kandilli Observatory Network[60]. The AFAD earthquake catalogue used in the statistical analysis starting in June 2020 is available from https://deprem.afad.gov.tr/event-catalog. The Güvercin et al.[25] catalogue presented in Fig. 1 can be obtained from https://doi.org/10.5281/zenodo.5220633. The enhanced seismicity catalogue produced in this study is available as a part of Supplementary Information.

## Code availability

None of the employed computer codes has been specifically developed for the purpose of this study. For estimation of the b-value and statistical properties, we employed our own MATLAB codes based on algorithms existing in the literature. For developing the enhanced seismicity catalogue, we adapted the methodologies employed for phase picking (https://github.com/AI4EPS/PhaseNet, last accessed 21/03/2023) and event association (https://github.com/AI4EPS/GaMMA, last accessed 21/03/2023), which are available in GitHub repositories. Single event earthquake localisation was performed with version 7.0.0 of the NLLoc software available at http://alomax.free.fr/nlloc/ and relative earthquake relocalization was performed with hypoDD available at https://www.ldeo.columbia.edu/-felixw/hypoDD.html.

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

## Acknowledgements

The authors acknowledge funding from the Helmholtz Association in the frame of the Young Investigators Group VH-NG-1232 (SAIDAN) (to D.B. and P.M.G.) and ERC Starting Grant -101076119 (QUAKEHUNTER) (to P.M.G.). This study was supported within the funding programme "Open Access Publikationskosten" Deutsche Forschungsgemeinschaft (DFG, German Research Foundation) - Project Number 491075472.

## Author contributions

P.M.G. led the project team and conceived the idea of the study. G.K., D.B. and P.M.G. performed formal analysis of the seismic data; M.B., D.A. and S.E. provided data access; G.C.B. contributed methods; P.M.G. and

G.K. processed the images; P.M.G., G.K., G.D. and F.C. wrote the original draft. D.B., G.K., G.D., P.M.G., G.C.B., F.C. and M.B. interpreted and discussed the results; G.D., G.C.B., G.K. P.M.G., F.C., D.A., S.E. and M.B. reviewed and finalised the manuscript.

## Funding

## Competing interests
The authors declare no competing interests.
