## [Peer Review File · Nature Communications]

Months-long seismicity transients preceding the 2023 MW 7.8 Kahramanmaraş earthquake, TürkiyeEditorial Note: This manuscript has been previously reviewed at another journal that is not operating a transparent peer review scheme. This document only contains reviewer comments and rebuttal letters for versions considered at *Nature Communications*.

REVIEWER COMMENTS

Reviewer #1 (Remarks to the Author):

I have reviewed this manuscript once before for Nature, and this revised version has been transferred to NCOMMS. The authors provide a very detailed response letter, itemizing how they have taken all of my comments and those of original reviewer #2 into consideration. Carefully documenting the seismicity leading up to this important earthquake sequence makes for an important contribution, and the manuscript is of high quality overall. The changes made in the revised version address my concerns, and I am supportive of publication of this version, presumably after addressing additional reviewers comments.

Roland Burgmann

Reviewer #2 (Remarks to the Author):

The present study suggests the seismicity transients starting in June 2022 prior to the occurrence of the 2023 Mw7.8 Kahramanmaras earthquake, Türkiye, using the event list released from AFAD. They identified an extended earthquake preparatory phase which consisted of isolated spatio-temporal clusters observed within 65 km of the future earthquake epicenter. They claim that an increase in the seismicity rates and a decrease in the b-value near the epicenter can be seen during the preceding nine years. It can be worthwhile and important to document and publish the seismicity transient described in the manuscript. However, I have a few concerns about the quality of the earthquake catalog and the unstable b-value variation (as described below). Furthermore, it is not easy to find a systematic difference between regular clusters or preparatory clusters that will culminate in a big earthquake, because the features of the preceding seismicity are weak, not so strong to define the “preparatory process” clearly before the mainshock rupture. It is very difficult to define the “preparatory process” based on current knowledge. Thus, reporting without mentioning the "preparation process" throughout the manuscript is less likely to mislead the seismological community, given the high impact of the journal.

1. I am concerned about the quality of the earthquake event catalog they used for the analysis because the catalog contains at least three clusters (C5, C6 and C8) caused by quarry blasts. The authors are not confident about the contamination of quarry blasts into the cluster C7. They identified blasts based on the narrow-magnitude band of the magnitude-frequency distributions, but I wonder if the explosion could still be included in the catalog. The depths of quarry blasts are generally close to the surface, the first-motion polarity of each vertical component is positive at all stations, and surface waveforms after P-wave arrival are dominant in the waveform of typical blasts, so it is not difficult to remove the quarry blasts as far as there are enough seismic stations. Why don't you use waveform-based approaches that

have been proposed by many previous studies aiming to discriminate blasts? Quarry blasts should be removed from the catalog before the elaborate analysis. Otherwise, the possible contamination of blasts may give a bias to the present results.

2. The authors enhanced the catalog using the machine learning-based model for only one month. Considering the purpose of this study, is the duration too short? At least, it is expected to analyze continuous data back to the time of the 2020 Elazik/Sivrice event to reveal the details of the transient earthquake activity lasting approximately 8 months before the mainshock rupture. The machine learning approach will provide a more complete earthquake catalog than the existing one.

3. The authors selected earthquake clusters located on and off the main fault trace near the hypocenter of the mainshock rupture. How do you split the events into these clusters? As far as I see Figure 2, many gray dots are hidden beneath some selected clusters or located nearby each cluster. Is each cluster selected arbitrarily? In general, it is not straightforward to choose each cluster from the complex earthquake distribution, but there is no clear explanation about it. Usually, we have applied some declustering algorithms in spatial and temporal domains, such as ETAS modeling and the rescaled space and time distance proposed by Zaliapin & Ben-Zion (2013).

4. The b-value within a 50 km radius shows a transient decrease and the subsequent quick recovery before the mainshock rupture. While the authors claim this change as evidence of low b-value, the temporal variations in b-values are quite scattered and unstable. It is thus challenging to persuade the low b-value near the mainshock epicenter. For example, a similar drop can be seen from the b-value in the distance range from 90 to 130 km in the middle of 2017 and 2019. How can you discriminate the significance of the temporal reduction in the b-value before the mainshock rupture?

5. Is there any possibility that the catalog constructed by the machine learning approach for the preceding one month includes quarry blasts (Figure 4)? Also, I am wondering how big the hypocenter errors in the catalog are. The hypocenters with major error ellipsoid half-axis < 10 km are colored gray in Figure 4, but the horizontal error of 10 km is still large even after applying the relative location procedure. The authors should select only more accurate hypocenters (< 3-5 km) because some non-well-relocated events or events contaminated by noises could be included in this recent catalog. In particular, the scattered distribution of the epicenters in Figure S7 appears to be an insufficiently constrained event, with the exception of a few clusters.

6. The index of magnitude distribution in time or interevent time statistics means just a temporal clustering of the seismicity. Seismic clustering often occurs without the subsequent earthquake, so it is unclear why the authors focus on the above indexes. Because seismicity by the ETAS model can produce significant clustering using these indexes. I wonder if the clustering feature is prominent regarding the preparatory process up to a large earthquake.

7. Regarding the three blasting clusters (C5, C6 and C8), can you confirm quarrying activity at each site by using other information (e.g, Blasting company location, Satellite image)?

8. The authors compare the observations with the laboratory experiments, but the AE activity in the

laboratory shows a rapid/dramatic increase in the generation rate. The observation of the increase in the seismicity rate is weak (Figure 3a) compared with the laboratory. So, it is necessary to temper the expression (e.g., L291: a substantial increase in the seismic event: not substantial) and add some discussions about the discrepancy.

REVIEWER COMMENTS

Reviewer #1 (Remarks to the Author):

I have reviewed this manuscript once before for Nature, and this revised version has been transferred to NCOMMS. The authors provide a very detailed response letter, itemizing how they have taken all of my comments and those of original reviewer #2 into consideration. Carefully documenting the seismicity leading up to this important earthquake sequence makes for an important contribution, and the manuscript is of high quality overall. The changes made in the revised version address my concerns, and *I am supportive of publication of this version*, presumably after addressing additional reviewers comments.

Roland Burgmann

Reply: We thank the Reviewer #1 for expressing support towards the publication of this paper, and for constructive comments in the previous round which have significantly improved the manuscript. The suggestions allowed to provide a more realistic perspective with respect to the possibilities of detecting a future earthquake preparatory process.

Reviewer #3

The present study suggests the seismicity transients starting in June 2022 prior to the occurrence of the 2023 Mw7.8 Kahramanmaras earthquake, Türkiye, using the event list released from AFAD. They identified an extended earthquake preparatory phase which consisted of isolated spatio-temporal clusters observed within 65 km of the future earthquake epicentre. They claim that an increase in the seismicity rates and a decrease in the b-value near the epicentre can be seen during the preceding nine years. It can be worthwhile and important to document and publish the seismicity transient described in the manuscript. However, I have a few concerns about the quality of the earthquake catalog and the unstable b-value variation (as described below).

Furthermore, it is not easy to find a systematic difference between regular clusters or preparatory clusters that will culminate in a big earthquake, because the features of the preceding seismicity are weak, not so strong to define the "preparatory process" clearly before the mainshock rupture. It is very difficult to define the "preparatory process" based on current knowledge. Thus, reporting without mentioning the "preparation process" throughout the manuscript is less likely to mislead the seismological community, given the high impact of the journal.

Reply: We appreciate the critical review of Reviewer#3. In this version of the manuscript and in the Rebuttal Letter, we have further documented that the quality of the employed catalogs is enough to ensure the validity of the presented results. Furthermore, we comment and describe the b-value variation.

As illustrated by the cited references in the manuscript, we are aware of the complexity and varying signatures of earthquake preparation processes, which can lead to ambiguous observations. We would like to emphasize that existing studies are still scarce and do not address several questions that are the focus of our study: a) Did the seismicity recorded before the 2023 Mw 7.8 event show signatures of an underlying accelerated deformation process culminating in the final extreme event? b) Which seismic characteristics could be used effectively to document an evolving process?, and c) What are similarities and differences between a simpler fault geometry (e.g. Izmit 1999 event) and a complex fault network as such at the East Anatolian Fault Zone?

Most importantly, we do not claim that the observed processes contain information on the size of the upcoming event. We agree with Reviewer#3 that the term preparatory process is somewhat loosely defined. Instead, we refer to laboratory results showing that accelerated deformation signified e.g. by enhanced seismic activity may occur during the seismic cycle but does not necessarily lead to a subsequent large event. We have modified the Introduction and Discussion sections of the manuscript accordingly to clarify the motivation and limitations of our study.

1. I am concerned about the quality of the earthquake event catalog they used for the analysis because the catalog contains at least three clusters (C5, C6 and C8) caused by quarry blasts. The authors are not confident about the contamination of quarry blasts into the cluster C7. They identified blasts based on the narrow-magnitude band of the magnitude-frequency distributions, but I wonder if the explosion could still be included in the catalog. The depths of quarry blasts are generally close to the surface, the first-motion polarity of each vertical component is positive at all stations, and surface waveforms after P-wave arrival are dominant in the waveform of typical blasts, so it is not difficult to remove the quarry blasts as far as there are enough seismic stations. Why don't you use waveform-based approaches that have been proposed by many previous studies aiming to discriminate blasts? *Quarry blasts should be removed from the catalog before the elaborate analysis.* Otherwise, the possible contamination of blasts may give a bias to the present results.

Reply: Thank you for this comment. We are aware that different methods are available to discriminate between earthquakes and quarry blasts, but ultimately, there is no methodology free from ambiguity. We discriminated between quarry blasts and earthquakes according to the available data and the following information (1) the temporal (daily) distribution of the events, which is uniform for earthquakes and highly inhomogeneous for quarries, (2) the magnitude-frequency distribution (MFD) and b-value of different clusters, where the MFD is much narrower and b-value is higher for quarries than for earthquakes, (3) the locations of events classified as quarry blasts by the KOERI seismic catalog, and, most importantly, (4) the observation of quarry locations using the Open Street Maps digital terrain map.

We used these criteria because other criteria suggested by the Reviewer are either unavailable to us or ambiguous. The hypocentral depth of the events included in the AFAD catalog is largely unconstrained, and hence this criterion cannot be used to identify quarries without a full reprocessing of earthquake catalog, which is beyond the scope of the present paper. Also, no information about polarities is available from the AFAD database. While it is possible to apply semi-automated or deep learning methods to infer polarity information, this cannot be done without a thorough analysis.

Event locations belonging to clusters Q5, Q6 and Q8 (renamed from C5, C6 and C8) are associated with quarry locations, and observations are further supported by (1), (2), and (3). We note that the criteria (1) and (2) were already presented in the paper, and we now added additional details regarding (3) and (4) in the manuscript. We reassessed cluster C7, that might still contain some component from quarry activity simply because there are some quarries in the area, thus point (4) is fulfilled. However, the temporal distribution over the day is flat, the MFD is substantially broader and b-value is substantially lower while comparing to the other quarry blast clusters, hence points (1) and (2) are not fulfilled. From the KOERI catalog, we found that events attributed as quarry blasts constitute up to 10% of the catalog for C7 in the time period of its highest activity. For comparison, clusters Q5 and Q6 contain 95% of quarry blasts, whereas Q8 contains at least 50% of quarry blasts. However, the remaining clusters are clearly earthquakes, containing on average less than 3% events attributable to quarry blasts. Therefore, we conclude that cluster C7 represents primarily seismic activity.

In the revised manuscript, we have added additional explanations regarding the criteria for cluster discrimination, as well as our interpretation of cluster C7. We also have added a comparison between clusters containing quarry blasts and seismic activity, as the one shown below to the Supplementary Figure S1.

Cover Letter Figure 1: Comparison of magnitude-frequency PDF for clusters containing seismic activity (solid black line), dominated by quarry blasts (dashed red line) and cluster C7 (orange line) for both KOERI and AFAD catalogs (left and right panels, respectively).

2. The authors enhanced the catalog using the machine learning-based model for only one month. Considering the purpose of this study, is the duration too short? At least, it is expected to analyze continuous data back to the time of the 2020 Elazik/Sivrice event to reveal the details of the transient earthquake activity lasting approximately 8 months before the mainshock rupture. The machine learning approach will provide a more complete earthquake catalog than the existing one.

Reply: The main reasoning behind the generation of our enhanced seismicity catalog was to illuminate with the highest resolution possible the regional seismicity during the *days* prior to the rupture. This was important because the AFAD seismicity catalog did not display immediate

foreshocks in the nucleation area, and we wanted to know whether additional seismicity was illuminated at lower detection thresholds.

Following Reviewer #1's recommendation, we extended our long-term analysis back to 2014 using the AFAD catalog and explored the time of the Elazig/Sivrice earthquake. Further extending the enhanced catalogue is currently not possible as continuous data still needs to be manually requested and downloaded in chunks of 24 hours, via the AFAD web interface. As of now this cannot be automated.

3. The authors selected earthquake clusters located on and off the main fault trace near the hypocentre of the mainshock rupture. How do you split the events into these clusters? As far as I see Figure 2, many grey dots are hidden beneath some selected clusters or located nearby each cluster. Is each cluster selected arbitrarily? In general, it is not straightforward to choose each cluster from the complex earthquake distribution, but there is no clear explanation about it. Usually, we have applied some declustering algorithms in spatial and temporal domains, such as ETAS modelling and the rescaled space and time distance proposed by Zaliapin & Ben-Zion (2013).

Reply: The clustering methods proposed by the Reviewer including ETAS modelling or the rescaled times and distance method by Zaliapin and Ben-Zion (2013) allow identification of background and triggered seismicity and/or sequences of foreshocks/mainshocks/aftershocks. A declustered catalog (background seismicity) might be more helpful to identify potential long term (several years) changes in the bulk damage of a region (e.g. Ben-Zion and Zaliapin, 2020, GJI). However, for tracking localization and delocalization processes in space and time, an entire catalog as employed in our study is more informative.

The primary purpose of our forming clusters was to highlight and analyse different spatial domains. Previously, we identified clusters based on visual inspection of the data and selecting the regions with larger event density. To reduce observer bias, we employed the density-based event clustering method DBSCAN to identify the clusters assuming $\epsilon=2.5\text{km}$ and a minimum number of events in the cluster $N=40$ (the latter was constrained to have enough data to calculate b -value). The objective DBSCAN method yielded a very similar clustering to the one previously employed, but it added one more (tectonic) cluster C9 that we now analysed as well. This resulted in only minor changes to the text and figures containing information on clusters. However no major changes to interpretation, discussion and conclusion parts were required.

4. The b -value within a 50 km radius shows a transient decrease and the subsequent quick recovery before the mainshock rupture. While the authors claim this change as evidence of low b -value, the temporal variations in b -values are quite scattered and unstable. It is thus challenging to persuade the low b -value near the mainshock epicentre. For example, a similar drop can be seen from the b -value in the distance range from 90 to 130 km in the middle of 2017 and 2019. How can you discriminate the significance of the temporal reduction in the b -value before the mainshock rupture?

Reply: Thank you for this comment. Indeed, we noticed other (smaller) fluctuations in the b -value. Based on the Reviewer's suggestions, we re-examined the data in the years 2017 and 2019

containing the b -value drops for the zones with $50 < R < 90$ km and $90 < R < 130$ km from the mainshock epicentre. In both years, the b -value drops are associated with delocalized seismicity in the western part of the region (see Figure below) that cannot be attributed to distinct clusters, but that still leads to transient changes in the magnitude-frequency distribution. This seismicity is associated with a fault system in the Osmaniye region (see Figure 1 from the manuscript). We also confirmed that the seismicity of C4 did not play an important role in the b -value drop for the 50-90 km zone, while C9 was inactive in both years. This highlights the importance of the concurrency of the spatial clustering and b -value drop before the mainshock, together with the proximity to the main fault trace, as observed for clusters C1 and C2.

We now comment on the temporal b -value drops in the manuscript text and in the caption of Figure 3 noting that these earlier b -value drops were not attributed to any spatio-temporal clusters but instead were related to a different fault system.

Cover Letter Figure 2: Map view of the seismicity in outer zones throughout the whole analyzed period (light green and blue points) and from the year 2017 (dark green and blue points). The seismicity from 2017 mostly occurred in the Osmaniye fault system, west of the future earthquake hypocenter, and represents scattered seismicity. This seismicity is responsible for the low b -value episodes observed in 2017 (as well as in 2019).

5. Is there any possibility that the catalog constructed by the machine learning approach for the preceding one month includes quarry blasts (Figure 4)?

Reply: It is almost impossible to be 100% certain of having a seismic catalog completely free from quarry blasts in this area. We are confident that the vast majority of our catalog events correspond to seismicity because of the following arguments: (1) The methodology to pick the P and S-waves (PhaseNet) is trained on earthquakes, not on quarry blasts, and as such, it will more effectively focus on detecting earthquakes. (2) During the analysed time in the enhanced catalog (01/01/2023

to 05/02/2023, both included), the KOERI catalog reported 3 quarry blasts out of 110 events in the study region (Lat and Lon [36-39]), corresponding to 2.7% of the total catalog. Only one quarry blast is less than 50 km away, near the city of Gaziantep. The other two are located in the eastern boundary of the study region, near the city of Sanliurfa.

In this version, we have additionally removed all events contained in the areas delimited by Q5, Q6 and Q8, which were identified areas of quarry blasts in the KOERI catalog. The number of events from the enhanced catalog located in the quarry areas was 26 out of the 1055. We have now added an explanation in the text about this, and updated the corresponding figures. The main results described in the paper did not change. As additional information, the density of hourly events calculated using a non-parameter kernel density distribution is provided below.

Cover Letter Figure 3: Probability density function (PDF) of the hours of event origin time calculated using the non-parametric kernel density approach confirming tectonic origin of the seismicity (cf. distributions for tectonic seismicity in Figure S1cd) from the extended catalog.

Also, I am wondering how big the hypocentre errors in the catalog are. The hypocenters with major error ellipsoid half-axis < 10 km are coloured grey in Figure 4, but the horizontal error of 10 km is still large even after applying the relative location procedure. The authors should select only more accurate hypocenters (< 3-5 km) because some non-well-relocated events or events contaminated by noises could be included in this recent catalog. In particular, the scattered distribution of the epicentres in Figure S7 appears to be an insufficiently constrained event, with the exception of a few clusters.

Reply: Thank you for the comment. We now clarified the characteristics of the different enhanced catalogs in the main manuscript, in addition to the information presented in the Materials and Methods section. For the investigated time interval, we obtained a single event NLLoc catalog that contains a total of 1055 events. Median errors of these events are 4.8 km, 5.5 km and 6.5 km in the x-, y- and z- directions, respectively. We did not perform a hypoDD relocation for the entire catalog, but only for the events associated with cluster C2 to better illuminate the activated fault structure and the spatio-temporal development of this activity (Fig. S8). After relative relocation employing hypoDD, a total of 86 events from initially 89 remained and are displayed in Fig. S8.

The formal median errors of these events in x-, y- and z-directions are 18 m, 35 m and 57 m, respectively.

Regarding the display of activity present in the enhanced single event NLLoc catalog in Fig. S7, we now follow the approach from Fig. 4 in the main manuscript and present the events with a major error ellipsoid half-axis > 10 km as grey dots. This highlights seismicity associated with C2 and other mapped faults as better constrained events are generally close to them and indicates that most of the scattered seismicity is not well constrained, as suggested by the reviewer. Taking a cut-off value of 5 km as suggested by the reviewer in contrast to the 10 km presented now also in Fig. S7 only leads to small changes in the overall event distribution (Cover Letter Figure 4).

Cover Letter Figure 4: Comparison of 10km (left) and 5 km (right) error cut-off for seismicity in the vicinity of the nucleation area of the first mainshock. Events with a with major error ellipsoid half-axis < 10 km and < 5 km, respectively, are color-coded according to their origin time and events with a with major error ellipsoid half-axis ≥ 10 km and ≥ 5 km, respectively, are shown as grey dots.

6. The index of magnitude distribution in time or interevent time statistics means just a temporal clustering of the seismicity. Seismic clustering often occurs without the subsequent earthquake, so it is unclear why the authors focus on the above indexes. Because seismicity by the ETAS model can produce significant clustering using these indexes. I wonder if the clustering feature is prominent regarding the preparatory process up to a large earthquake.

Reply: In this study we focused on the properties of clustered seismicity. Indeed, the interevent time ratio statistics show temporal clustering for the earthquake clusters, suggesting interaction between events or stress transfer. However, spatial clustering may occur for earthquakes without temporal clustering suggesting, e.g. the lack of interaction between events (e.g. Kwiatek et al., 2022) or the presence of external aseismic forcing such as slow slip or fluids (e.g. Martínez-Garzón et al., 2019; Becker et al., 2023). In contrast, the interevent time ratio showed degenerated statistics indicating repetitive events for some quarry-polluted clusters, which might be interesting to show as an additional indicator of seismic catalog pollution with mine blasts. Finally, the magnitude correlation analysis highlighted the peculiar behaviour of the C1 cluster, being the only one that shows signatures of cascading acceleration/deceleration. As stress transfer / earthquake interaction

and accelerating seismic release is important while discussing the preparatory processes, we wish to keep these figures in. We modified small portions of the text and caption of Figure S6 to highlight the observations related to these statistics.

7. Regarding the three blasting clusters (C5, C6 and C8), can you confirm quarrying activity at each site by using other information (e.g., Blasting company location, Satellite image)?

Reply: Yes, we can confirm that. Specifically, we performed visual inspections using Open Street Maps and available map overlays. Now, clusters are named Q5, Q6 and Q8 to clearly distinguish them from seismicity clusters.

Cover letter Figure 5: Example of quarry representation in OpenStreetMaps located NE from the city of Gaziantep correlating with epicentral locations of events from cluster Q6 (left: broader region with the city of Gaziantep located in the SW part of the map; right: zoom-in of the central part showing quarry polygon). Quarry polygons were queried from the service and compared with surface distribution of seismicity.

8. The authors compare the observations with the laboratory experiments, but the AE activity in the laboratory shows a rapid/dramatic increase in the generation rate. *The observation of the increase in the seismicity rate is weak (Figure 3a) compared with the laboratory.* So, it is necessary to temper the expression (e.g., L291: a substantial increase in the seismic event: not substantial) and add some discussions about the discrepancy.

Reply: In laboratory experiments the activity increase prior to failure depends on boundary conditions such as effective pressure, loading rate, fault roughness etc. Rapid increase in the seismicity rate (e.g. exponential) is observed in laboratory fracture experiments, for example, on intact samples, or sometimes in friction (stick-slip) experiments on smooth faults. However, for stick-slip experiments on rough faults, the activity rate increase may be more modest at high levels of stress (e.g. Dresen et al., 2020). Rough lab faults may likely be a better approximation of the preparatory processes in complex fault zones in nature, as in the case of the Kahramanmaraş earthquake. Following Dresen et al., (2020) and Kwiatek et al., (2023), rough faults are characterized by initially growing and ultimately varying seismicity rates during the later parts of the preparatory phase, in which the asperities activate, superimpose and sometimes interact, collectively preparing the wider region for a system-size earthquake. Indeed, in the previous

version we were not specific enough regarding boundary conditions of the lab experiments that we meant to be comparable with our observations. We therefore extended the discussion to better explain the similarities between laboratory experiments and our observations.

References:

Becker, D., Martínez-Garzón, P., Wollin, C., Kılıç, T., & Bohnhoff, M. (2023). Variation of Fault Creep Along the Overdue Istanbul-Marmara Seismic Gap in NW Türkiye. *Geophysical Research Letters*, 50(6), e2022GL101471. <https://doi.org/10.1029/2022GL101471>

Dresen, G., Kwiatek, G., Goebel, T., & Ben-Zion, Y. (2020). Seismic and Aseismic Preparatory Processes Before Large Stick–Slip Failure. *Pure and Applied Geophysics*, 177(12), 5741–5760. <https://doi.org/10.1007/s00024-020-02605-x>

Kwiatek, G., P. Martínez-Garzón, J. Davidsen, P. Malin, A. Karjalainen, M. Bohnhoff, and G. Dresen (2022). Limited Earthquake Interaction During a Geothermal Hydraulic Stimulation in Helsinki, Finland, *Journal of Geophysical Research: Solid Earth* **127**, no. 9, e2022JB024354, doi [10.1029/2022JB024354](https://doi.org/10.1029/2022JB024354).

Kwiatek, G., P. Martínez-Garzón, T. Goebel, M. Bohnhoff, Y. Ben-Zion, and G. Dresen (2023). Complex multi-scale preparatory processes of stick-slip events on rough laboratory faults, doi [10.22541/essoar.169447455.58529925/v1](https://doi.org/10.22541/essoar.169447455.58529925/v1).

Martínez-Garzón, P., Ben-Zion, Y., Zaliapin, I., & Bohnhoff, M. (2019). Seismic clustering in the Sea of Marmara: Implications for monitoring earthquake processes. *Tectonophysics*, 768, 228176. <https://doi.org/10.1016/j.tecto.2019.228176>

REVIEWERS' COMMENTS

Reviewer #2 (Remarks to the Author):

The authors responded effectively to my comments and subsequently revised certain sections of the manuscript. I believe the manuscript is now ready for publication.